# Cryo-EM single particle analysis with the Volta phase plate

**Radostin Danev\*, Wolfgang Baumeister**

Department of Molecular Structural Biology, Max Planck Institute of Biochemistry, Martinsried, Germany

**Abstract** We present a method for in-focus data acquisition with a phase plate that enables near-atomic resolution single particle reconstructions. Accurate focusing is the determining factor for obtaining high quality data. A double-area focusing strategy was implemented in order to achieve the required precision. With this approach we obtained a 3.2 Å resolution reconstruction of the *Thermoplasma acidophilum* 20S proteasome. The phase plate matches or slightly exceeds the performance of the conventional defocus approach. Spherical aberration becomes a limiting factor for achieving resolutions below 3 Å with in-focus phase plate images. The phase plate could enable single particle analysis of challenging samples in terms of small size, heterogeneity and flexibility that are difficult to solve by the conventional defocus approach.

## Introduction

Phase plates are one of the technologies holding promise for future performance improvements in cryo-electron microscopy (cryo-EM). Direct electron detectors already changed the prospects of the cryo-EM field (*Henderson, 2015*) with atomic resolution structures becoming almost routine (*Campbell et al., 2015*; *Bartesaghi et al., 2015*). Phase plates improve the image contrast and allow in-focus data acquisition which, in theory, will result in increase of the signal-to-noise ratio across the entire frequency spectrum (*Glaeser, 2013*). This could enable structural investigations of 'difficult' samples, such as small, heterogeneous and/or flexible molecules or complexes (*Hall et al., 2011*). Until now, however, various practical problems and performance issues have prevented the acquisition of high-resolution datasets with a phase plate (*Danev et al., 2009*). Here we demonstrate that with correct use, in particular accurate focusing, one can achieve near-atomic resolutions.

The phase plate is a device that produces phase contrast by introducing a phase shift between the scattered and unscattered waves at a diffraction plane inside the microscope. Phase contrast denotes that phase variations of the electron wave caused by the sample will be transformed into amplitude variations at the camera thus enabling phase observation. Ideally, the phase shift must be $\pi/2$ to realize the so-called Zernike phase contrast (*Danev and Nagayama, 2001*). In practice, however, a satisfactory phase contrast performance can be obtained within a range of phase shift values ($\pi/4 \sim 3\pi/4$) (*Danev and Nagayama, 2011*). There are other ways to produce phase contrast, the most common is acquiring images slightly out of focus, also known as defocus phase contrast. This method is the de facto standard in transmission electron microscopy (TEM) but it has the disadvantage of low overall contrast because of poor performance at low spatial frequencies, i.e. large specimen features are not well reproduced in the image.

Phase plates for TEM have been in development for more than 15 years with the thin film Zernike phase being one of the most promising candidates (*Glaeser, 2013*). It consists of a thin material film, typically amorphous carbon, with thickness selected for $\pi/2$ phase shift at the operating voltage of the microscope (~22 nm for 100 kV; ~31 nm for 300 kV) and a small (~1 µm) hole in the center for the central beam of unscattered electrons (*Danev and Nagayama, 2001*). Nevertheless, the Zrnike

\*For correspondence: danev@
biochem.mpg.de

**Competing interest:** See
page 12

**Reviewing editor:** Sjors HW
Scheres, Medical Research
Council, United Kingdom

**eLife digest** One way of investigating how proteins and other biological molecules work is to look at their structure. Light microscopes cannot produce detailed enough images to fully reveal these structures, and so a technique called cryo-electron microscopy is often used instead. In this technique, a biological sample is frozen to the temperature of liquid nitrogen and a beam of electrons is fired at it to create an image. By taking many of these images and then subjecting them to computer processing it is possible to reconstruct the three-dimensional structure of the molecule.

Frozen biological samples are essentially transparent to the electron beam used in an electron microscope. To view samples, researchers therefore use a method called phase contrast, which relies on a property of the electron beam (called its phase) changing as the beam passes through the sample. The traditional "defocus" method of producing phase contrast from electron microscopy relies on processing a series of slightly out-of-focus images of the sample.

Phase plates are add-on devices that are commonly used in light microscopes to produce phase contrast. For many years now, attempts have been made to produce a working phase plate for electron microscopes. However, an effective plate, called the Volta phase plate, has only recently been developed.

Danev and Baumeister have now evaluated how well the Volta phase plate performs during the analysis of a single, relatively large protein. This molecule is considered 'easy' to analyze using cryo-electron microscopy as relatively few microscopic images need to be recorded to solve the protein's structure. Danev and Baumeister found that the Volta phase plate matched or slightly exceeded the performance of the traditional defocus method of producing phase contrast, depending on how many images were used to analyze the protein. This is the first time that a phase plate has matched the performance of the defocus method.

A future challenge will be to make the experimental procedures and the software involved in using the Volta phase plate more user-friendly. The phase plate also needs to be tested with more 'difficult' samples, such as small proteins and samples whose structure could not be established using the defocus method of producing phase contrast.

phase plate has some practical issues, such as a short usable lifespan, produces fringe artifacts in the images, is difficult to use and almost impossible to automate because the central diffraction beam must be positioned precisely in the middle of the small central hole (*Danev et al., 2009*). There are a few examples in the literature of single particle analysis (SPA) with the Zernike phase plate but in all of them the resolution was limited to about one nanometer (*Danev and Nagayama, 2008*; *Murata et al., 2010*).

The Volta phase plate (VPP) was developed recently as a successor to the thin film Zernike phase plate (*Danev et al., 2014*). It solves most of the issues, in particular, the VPP has a virtually unlimited life, does not produce fringe artifacts and does not require precise centering. There are already a few application examples of the VPP in cryo-tomography (*Fukuda et al., 2015*; *Asano et al., 2015*; *Mahamid et al., 2016*) and single particle analysis (*Khoshouei et al., 2016*) that demonstrate its benefits.

The VPP consists of a thin (~10 nm) continuous amorphous carbon film which is constantly heated to ~200°C to prevent beam-induced contamination. The phase shift is created on-the-fly through the interaction of the central diffraction beam with the carbon film (*Danev et al., 2014*). This mitigates the requirement for precise centering of the phase plate but also gives rise to the only drawback of the VPP – the phase shift is not constant and increases proportionally to the accumulated dose. The evolving phase shift is not a performance limiting factor but rather a parameter that has to be kept in mind when designing data acquisition strategies. Nevertheless, the ability to create phase shift at any given place on the phase plate film is a notable practical advantage. A wide open area of the film provides multiple virtual phase plates by just moving from one position to the next. The distance between the positions must be large enough (typically ~20 μm apart) to prevent the previous phase shift spot from interfering with spatial frequencies below the Nyquist frequency of the detector. Having the ability to move arbitrarily on the phase plate and create phase shift opens the VPP to

automation. This is an important practical factor in sync with recent trends towards unattended 24/7 data collection. All datasets in this work were collected automatically.

## Results and discussion

Our initial SPA trials with the VPP were limited to resolutions (~8A, unpublished data) similar to those already reported with the Zernike phase plate. In those trials we used software and data acquisition schemes developed for conventional defocus phase contrast transmission electron microscopy (CTEM). Such schemes place low priority on focusing accuracy because with CTEM the defocus is accurately measured by contrast transfer function (CTF) fitting during data processing and some defocus variation is actually desirable and beneficial for the reconstruction (*Cheng et al., 2015*). As shown later, using phase plates in-focus requires very accurate focusing. Therefore, we had to develop and apply a new data acquisition scheme.

In general, there are two defocus strategies for SPA data acquisition with a phase plate. The first one involves collecting data in a similar fashion as in CTEM, i.e. with an intentionally applied defocus. The CTF must then be fitted and corrected during data processing. The advantage of this approach is that precise focusing in not required and some defocus variation is actually desirable. Not having to focus accurately also speeds up the acquisition. The disadvantage is that the only benefit from using a phase plate is contract increase for the very low spatial frequencies whereas the remainder of the frequency spectrum has multiple CTF zeroes, like in CTEM. Also, because of the phase plate, fitting the CTF requires an additional phase shift parameter which may reduce the accuracy of the fit. The second and ideal SPA acquisition scheme with a phase plate is to collect data in-focus. The advantage of this approach is that there are no CTF zeroes across the frequency spectrum and there is no need to fit and correct the CTF. However, this approach is more demanding on the experiment in that it requires very accurate focusing. Also, if there are small random defocus errors they cannot be corrected during data processing because fitting the CTF without at least a few detectable CTF zeroes is practically impossible. In our initial SPA trials with the VPP we also tried the defocused approach but without much success (unpublished data), mainly because of the lack of phase shift support in the single particle reconstruction software. Future software developments could change the status quo and make the defocused approach more attractive, but so far we have had more success with the in-focus approach (*Khoshouei et al., 2016*).

*Figure 1* illustrates the effect of defocus on the expected resolution for in-focus data collection with a phase plate. We use the term 'in-focus' also for small (<100 nm) defocus values which are used to counteract the effect of spherical aberration and extend the resolution range by moving the first CTF zero to higher spatial frequencies. *Figure 1A* shows simulated CTFs for a $\pi/2$ phase plate at three defocus values. The figure illustrates what would be an acceptable defocus range for a 4 Å resolution SPA reconstruction. The resolution criterion we used is CTF amplitude dropping below 0.5, i.e. |CTF|=0.5. The zero defocus curve (dashed line) illustrates the effect of spherical aberration on the CTF. The CTF crosses the 0.5 level at a resolution lower than 4 Å. Small amounts of underfocus (defocus < 0) counter the effect of spherical aberration and extend the region of good information transfer to higher resolutions. The red curve shows the minimum defocus ($-7$ nm) required to reach 4 Å resolution according to the |CTF|=0.5 criterion. Increasing the defocus improves the resolution further but at higher defocus values the initially flat part of the CTF starts to develop a dip, as illustrated by the blue curve (defocus -60 nm). This imposes a limit on the maximum defocus value.

Based on the |CTF|=0.5 criterion we calculated the usable defocus range as a function of the resolution. The result is shown in *Figure 1B*. The shaded areas are 'prohibited' in terms of CTF performance. The one on the right is caused by the CTF amplitude crossing the 0.5 point on its way to the first CTF zero, similar to the red curve in *Figure 1A*. The shaded area on the left is due to the dip in the flat region of the CTF exhibited by the blue curve in *Figure 1A*. This dip imposes a limit for all spatial frequencies above the point where it touches the 0.5 level which is the reason for the vertical wall-like defocus cutoff for a range of resolutions in *Figure 1B*. The white region between the shaded areas in *Figure 1B* is the usable defocus range (horizontal axis) for achieving a given resolution (vertical axis). For higher resolutions (lower numerical values) the defocus range gets narrower and has a cutoff point at about 2.7 Å. Achieving resolutions below that point will require reduction or complete elimination of the spherical aberration (Cs correction).

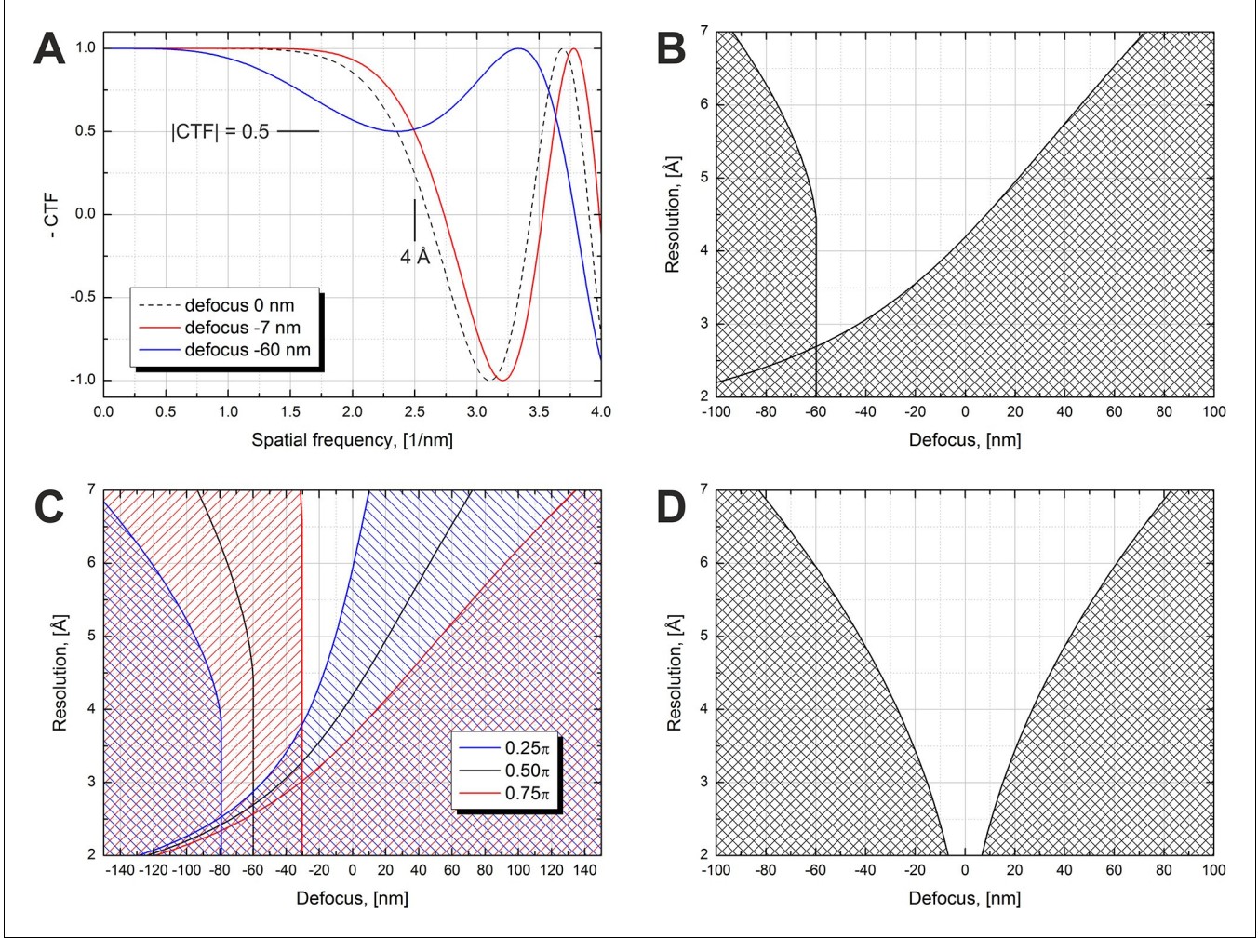

**Figure 1.** Volta phase plate CTF examples and allowed defocus ranges versus resolution. (A) Illustration of CTFs at defocus values that limit the resolution to 4 Å according to a |CTF|=0.5 criterion. (B) Defocus limits versus resolution according to the |CTF|=0.5 criterion for a $\pi/2$ phase plate and 2.7 mm spherical aberration. The shaded areas are 'prohibited' in a sense that for those defocus values the CTF amplitude drops below 0.5 at a resolution lower than the value on the y-axis. (C) Same as (B) but for three different phase shift values. (D) Same as (B) but for a Cs-corrected microscope (0 mm spherical aberration).

The phase shift of the VPP is not constant and increases with the accumulated dose on the phase plate (see *Figure 3* below) (*Danev et al., 2014*, *Figure 1*). It has a rapid onset in the beginning followed by a gradual increase. For single particle data acquisition this would mean that the first images taken after the phase plate is inserted will have a phase shift below $\pi/2$ and later images may have a phase shift above that value. *Figure 1C* illustrates the effect of phase shift on the usable defocus range and resolution. For lower phase shifts (blue area) the defocus ranges are shifted towards more defocus and there is a slight improvement in the absolute resolution cutoff point (~2.6 Å). Higher phase shifts (red area) require less defocus but have a worse resolution cutoff (~3.0 Å).

*Figure 1D* shows an ideal case of a $\pi/2$ phase plate on a Cs-corrected microscope (zero spherical aberration). There is no resolution cutoff but the allowed defocus range gets progressively narrower for higher resolutions. A variable phase shift will move the shaded areas to the left or right, similar to the behavior in *Figure 1C*, but in order to avoid clutter the effect is not shown in the figure.

The graphs in *Figure 1* demonstrate the strict requirements on the focusing accuracy for in-focus data acquisition with a phase plate. In order to achieve the required nanometer-level focusing precision we implemented a double focusing area acquisition scheme. *Figure 2A* shows an illustration of

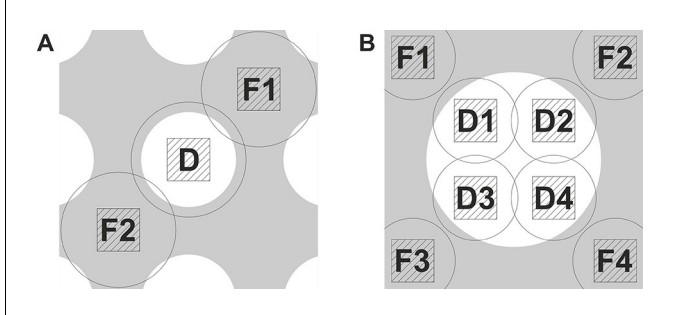

**Figure 2.** Data acquisition schemes that enable precise focusing superposed on a holey support film. (**A**) Two focusing areas, **F1** and **F2**, on opposite sides of the data acquisition position **D** are used to perform a linear interpolation of the measured defocus. (**B**) Multiple focusing areas **F1** to **F4** around a support film hole can be used to interpolate the defocus and acquire multiple data images (**D1** to **D4**) per hole.

the scheme superposed on a holey support film. The defocus was measured on opposite sides, areas F1 and F2, of the data acquisition area D and the average (linear interpolation) was used to correct the focus. Such measurement minimizes errors due to local slant of the support film. We measured up to 400 nm defocus difference across the two sides (3.0 um distance) of holes on Quantifoil R 1.2/ 1.3 holey carbon grids. This would mean that a single focusing area acquisition scheme will have a defocus error of up to 200 nm which is substantial compared to the values in *Figure 1*. That provides a plausible explanation for the lower resolution (~8 Å) reconstructions we obtained when using a conventional single focusing area scheme. To improve the accuracy even further the double area focusing was iterated 2 or 3 times depending on how fast it was converging (we typically aimed at +/− 10 nm from the target defocus). In order to prevent errors due to objective lens hysteresis the defocus was measured without applying a defocus offset during the measurement. The phase plate provides enough contrast for beam-tilt focus measurements even close to focus.

*Figure 2B* shows another possible acquisition scheme. Holey support films with larger holes permit collection of several data images (D1 to D4) within a single hole. By measuring the defocus at several points around the hole (F1 to F4) it will be possible to calculate a local plane or curvature model of the support film and apply the necessary focus correction for each acquisition area.

Spherical aberration influences not only the optimal defocus but also the measured defocus. The most commonly used method to measure defocus is the beam tilt method (*Koster et al., 1987*). It uses the shift between images taken with tilted beams to calculate the defocus. The effect of spherical aberration on the measurement is usually ignored because its contribution is small compared to the typical defocus values used in CTEM. Spherical aberration has an effect equal to a defocus of $\triangle z$ $=C_S\beta^2$, where $C_S$ is the spherical aberraton coefficient and $\beta$ is the beam tilt angle (*Koster et al., 1987*, equation 2). A commonly used tilt angle value is 5 mrad but in our experiments we used 10 mrad in order to improve the sensitivity of the measurement and to move the spots created on the VPP by the tilted beams further away from the central spot. On an FEI Titan Krios microscope (FEI, Hillsboro, OR) a 10 mrad beam tilt places the beam spot at a position corresponding to $k = \beta /\lambda =$ 0.01 rad / 0.002 nm = 5 nm$^{-1}$ spatial frequency or 2 Å resolution ($\lambda$ is the electron wavelength). This prevents the phase shifting spots created on the phase plate by the tilted beams from disturbing the CTF in the usable frequency range. The spots themselves are beneficial for the focusing because they act as phase plates themselves and increase the contrast of the tilted beam images, thus facilitating the measurement.

The offset in the measured defocus caused by spherical aberration with 10 mrad beam tilt on an FEI Titan Krios is $\triangle z =C_S\beta^2$ = 2.7 mm x 0.01$^2$ rad$^2$ = 270 nm. This again is a significant amount of defocus compared to the defocus limits in *Figure 1*. We used SerialEM software (*Mastronarde, 2005*) for the data acquisition which does not take into account the effect of spherical aberration on focus measurements. Therefore, we had to adjust the target defocus by the above offset, i.e. to get -20 nm actual defocus we had to set the desired defocus at 270 − 20 = 250 nm in the software.

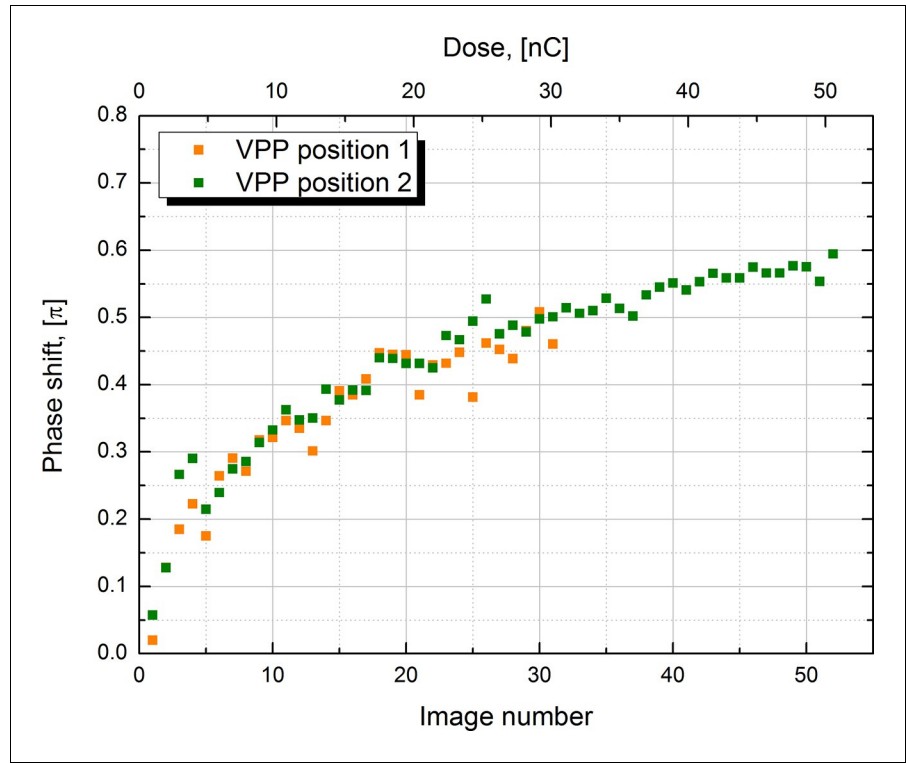

**Figure 3.** Measured phase shift of the VPP as a function of the image number/total dose on the phase plate. The measurement was performed at two consecutive positions on the phase plate.

In order to characterize the behavior of the VPP we measured the phase shift under the same experimental conditions as those used to collect the 20S proteasome datasets. Series of images were recorded automatically on carbon film parts of the sample with -1.5 µm defocus to facilitate phase shift measurement through CTF fitting. The CTFs of the image series were fitted with the latest version of ctffind4 (*Rohou and Grigorieff, 2015*) which supports phase shift. The results from two image series recorded at two consecutive positions on the VPP are plotted in *Figure 3*. Both series show very similar behavior indicating that there are no significant variations in performance between neighboring positions on the phase plate. The phase shift has a rapid onset in the beginning of the series but the rate of its development slows down as the series progress. The results reproduce well the previously reported VPP behavior (*Danev et al., 2014*, *Figure 1B*). During single particle data acquisition we try to prevent the phase shift from going much over $\pi/2$ by periodically moving to a new position on the phase plate (depending on the experimental conditions, every 25 to 50 images). Changing the phase plate position more often increases the risk of astigmatism due to variations in the phase plate film quality. Therefore, it is desirable to have a phase plate with a slower rate of phase shift development. In practice the rate can be reduced by increasing the phase plate heating temperature (*Danev et al., 2014*) but in the current VPP generation such control is quite limited by the specifications of the heater.

Using the acquisition scheme shown in *Figure 2A* we acquired two in-focus VPP datasets of the *Thermoplasma acidophilum* 20S proteasome, one at $-20$ nm and one at $-50$ nm defocus. From the same grid square we also acquired a CTEM dataset with a defocus range of $-0.8$ to $-1.7$ um. The datasets were acquired automatically using SerialEM macros (*Mastronarde, 2005*). The phase plate was automatically moved to a new area every ~1 hr (every ~27 images) to prevent too much phase shift buildup. Representative images from the datasets are shown in *Figure 4*. The VPP image (*Figure 4A*) has higher overall contrast compared to the CTEM image (*Figure 4C*) because of the improved low frequency transfer. The high contrast is very helpful during the particle picking process. The 2D Fourier transforms shown in *Figures 4B and C* illustrate the CTF differences between in-focus and defocused acquisitions. The VPP (*Figure 4B*) has a uniform transfer without visible

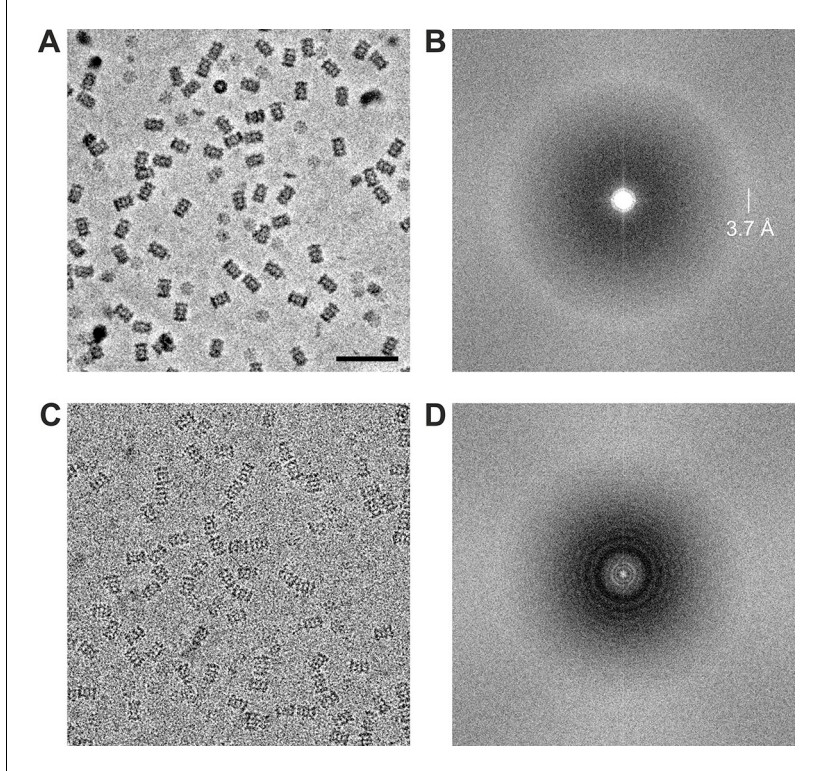

**Figure 4.** Representative images from 20S proteasome datasets acquired with and without a phase plate. (**A**) In-focus image acquired with the Volta phase plate. (**B**) Power spectrum of the image in (**A**). The presence of the amorphous ice ring at 3.7 Å indicates that there is good information transfer to at least that spatial frequency. (**C**) Conventional defocus image at −1.6 μm defocus. (**D**) Power spectrum of the image in (**C**) showing CTF Thon rings. Scale bar: 50 nm.

zeroes. The first CTF zero is beyond the 3.7 Å amorphous ice ring and is not detectable in the transform. The spectrum of the CTEM image (*Figure 4D*) shows characteristic CTF oscillations with multiple zeroes. Both power spectra exhibit a noticeable amplitude decrease in their central region which is a consequence of the relatively high dose rate on the detector (~9 e⁻/pixel/s). Higher dose rates increase the coincidence loss during electron counting and lead to amplitude reduction at low spatial frequencies but have little effect on the spectral signal-to-noise ratio (*Li et al., 2013b*).

An isosurface representation of the reconstructed 3D map from the -20 nm defocus VPP dataset is shown in *Figure 5A*. The reconstruction is based on 13,469 particles selected after a 3D classification step from an initial dataset of 35,469 particles. *Figure 5B* shows a part of the map superposed on an α-helix from the β subunit of a 20S atomic model (PDB 3J9I) and demonstrates the presence of side chain densities. *Figure 5C* contains plots of the Fourier shell correlations (FSC) calculated by the internal 'gold standard' procedure in Relion (*Scheres, 2012*) (blue curve) and versus an external 2.8 Å resolution density map (EMD-6287, *Campbell et al., 2015*). Both criteria give an identical resolution estimate of 3.2 Å at 0.143 level for the 'gold standard' FSC and at 0.5 level for the external map FSC. This is the highest resolution phase plate single particle reconstruction reported to date.

*Figure 6A* contains plots of resolution versus number of particles for various datasets, collected with and without a VPP. The solid lines represent the two main VPP and CTEM datasets collected on two halves of the same grid square. For each point a complete 3D refinement run was performed in Relion using a 60 Å low-pass filtered initial model to avoid model bias. The number of particles was varied by extracting random subsets of particles. The resolutions were calculated based on the 0.5 level FSC versus the external map (EMD-6287). Both the VPP and the CTEM show a gradual improvement of resolution as the number of particles increases with the performance being virtually identical between the two techniques. The conventional dataset contained more micrographs (293

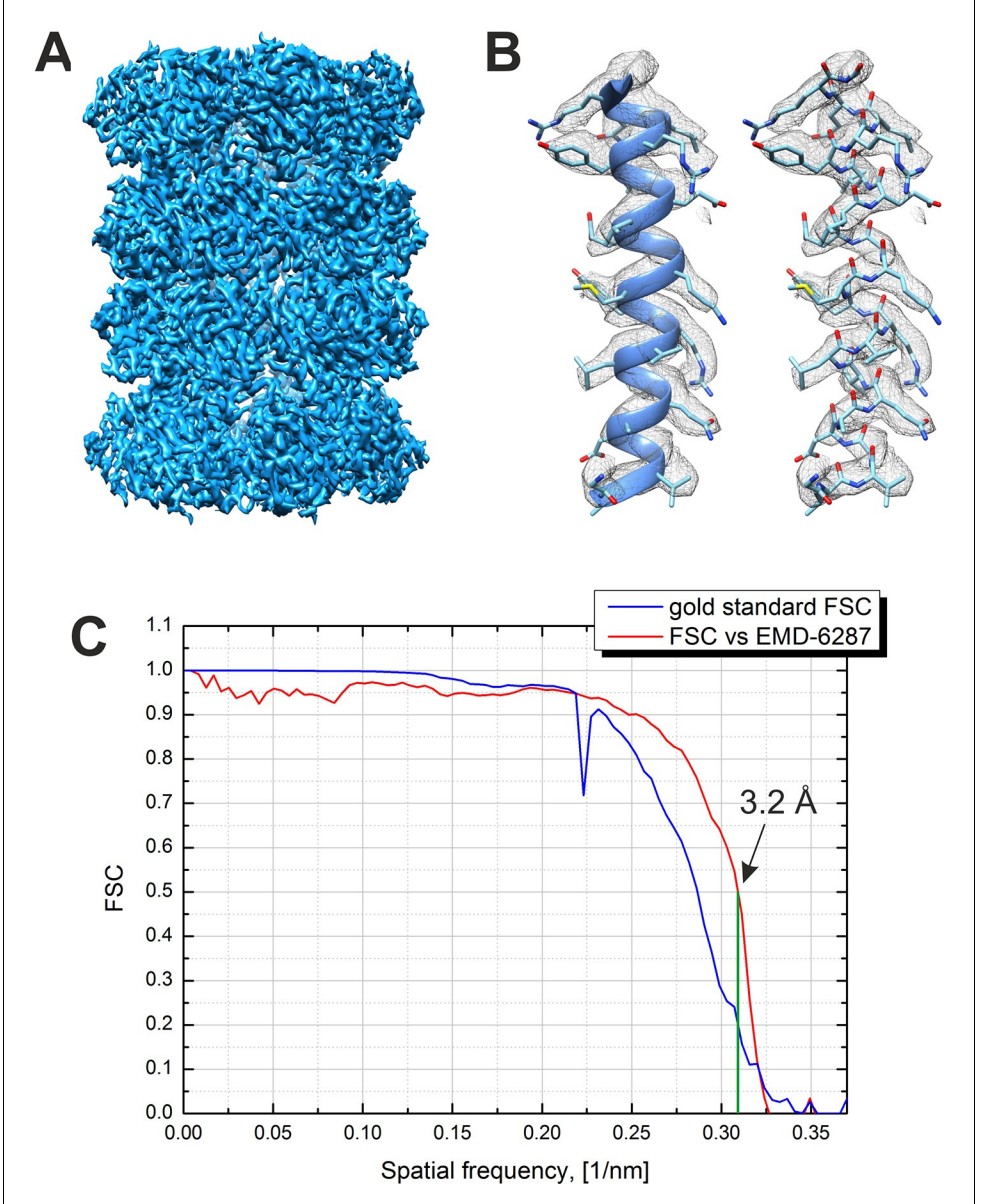

**Figure 5.** Result from the 3D reconstruction of 20S proteasome from an in-focus dataset acquired with the Volta phase plate. (**A**) An isosurface representation of the density map. (**B**) A ribbon and a stick representations of an α-helical segment from the β subunit docked into the density map demonstrating the presence of sidechain densities. (**C**) Fourier shell correlation (FSC) curves from the Relion software's internal 'gold standard' and from a comparison with an external 2.8 Å density map (EMD-6287). Both criteria indicate a resolution of 3.2 Å at a 0.143 level for the 'gold standard' FSC and at a 0.5 level for the external map. The Nyquist frequency is at 2.7 Å (0.37 1/nm).

vs 158) hence it had more particles (after 3D classification, 35,717 vs 13,469). With more than twice the number of particles the CTEM dataset reached a resolution of 3.1 Å. The measured B-factors in Relion were very similar between the two datasets: 119 for CTEM and 123 for VPP.

In order to estimate the B-factors from the resolution versus number of particles data we plotted the logarithm of the number of particles as a function of the squared reciprocal resolution (*Figure 6B*). The B-factor can then be estimated by a linear fit and is equal to twice the slope of the line (*Rosenthal and Henderson, 2003*, Figure 11). We noticed two approximately linear regions in both the CTEM and the VPP data. The region with particle numbers between 500 and 2000 has a B-

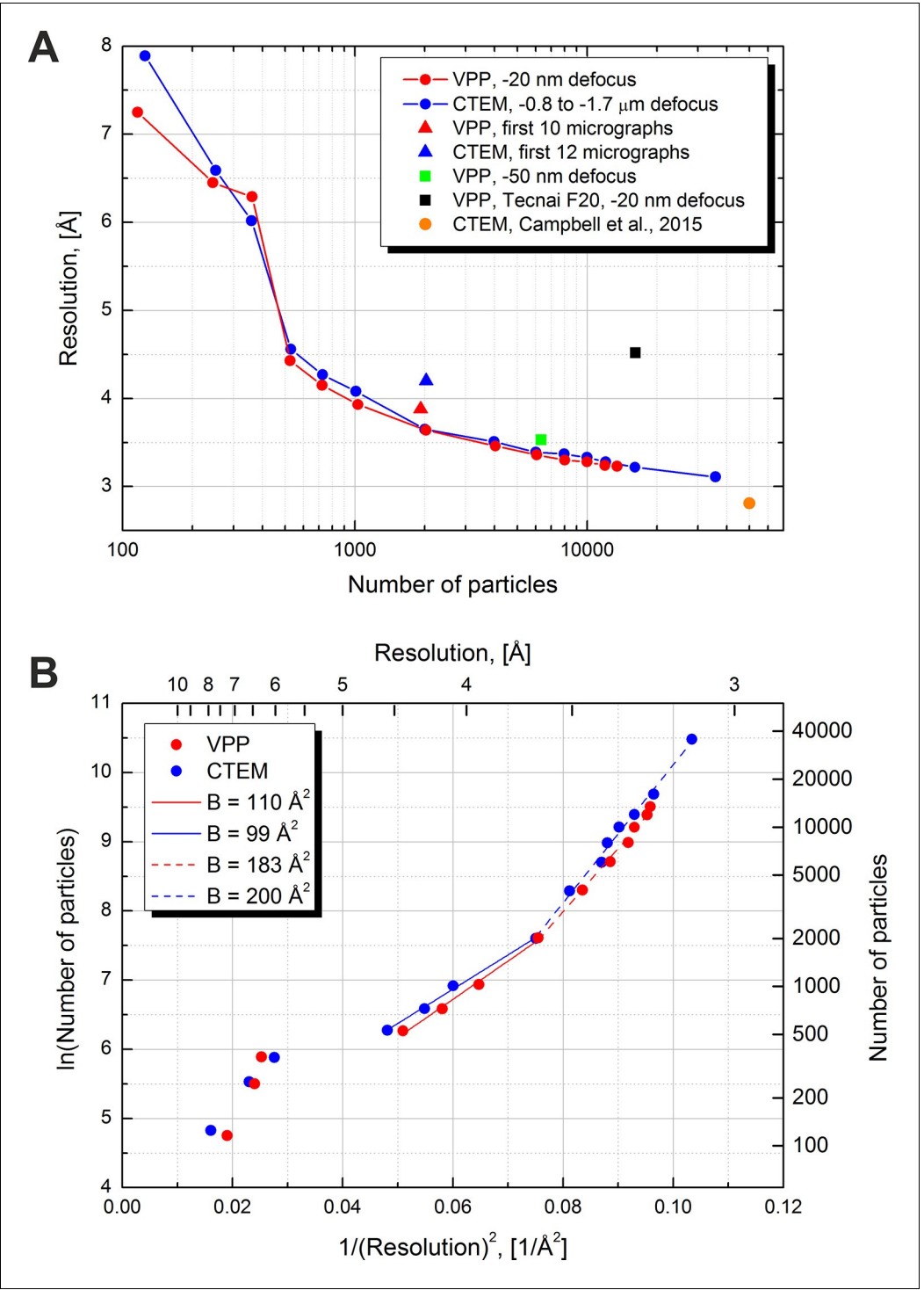

**Figure 6.** Resolution versus number of particles and B-factor estimations. (**A**) Resolution versus number of particles for several datasets. The resolutions were calculated based on the 0.5 FSC level versus an external 2.8 Å density map (EMD-6287). For two of the datasets (red and blue lines) the number of particles was varied artificially by using random subsets of particles. (**B**) Logarithm of the number of particles versus the squared reciprocal resolution plots of the two main datasets in (**A**). The legend contains B-factors estimated by linear fits of two data regions.

factor of ~100 and resolutions above 3.6 Å (solid lines in *Figure 6B*). The higher resolution region with particle numbers above 2000 has a B-factor of ~200 (dashed lines in *Figure 6B*).

*Figure 6A* also contains individual points for other datasets. A VPP dataset collected at -50 nm defocus and consisting of 6340 particles reached a resolution of 3.5 Å which is marginally (0.2 Å) worse than the same number of particles from the -20 nm defocus VPP dataset. According to the values of the phase shift in *Figure 3* and comparing them with the curves in *Figure 1C* the -50 nm defocus dataset should have matched or exceeded the performance of the -20 nm defocus dataset. This would suggest that either there was a small systematic focusing error or a variation in the phase plate performance. The black square in *Figure 6A* is for a VPP dataset collected on an FEI Tecnai F20 microscope equipped with an FEI Falcon II (FEI, Hillsboro, OR) direct detector camera. This microscope operates at 200 kV and uses a side-entry cryo-holder. It is inferior to the FEI Titan Krios microscope in both optical performance and specimen stage stability but the obtained resolution of 4.5 Å is still respectable for such a system. The orange circle shows the result from *Campbell et al. (2015)*, and is at 2.8 Å with 49,954 particles. In that work a much larger dataset of ~1000 micrographs was collected which was then subjected to CTF screening and only 193 micrographs (~19% of the original dataset) were selected for data processing. We performed only visual pre-screening of the datasets and excluded micrographs that had obvious faults, such as broken ice, large contaminants or duplicate exposures. Consequently, we used ~80% of the original micrographs for data processing. The smaller number of micrographs combined with the fact that our data was not collected in 'super resolution' mode of the K2 camera (the data in *Campbell et al, 2015*, was collected in 'super resolution' mode) could explain the slightly lower resolution of our reconstructions.

To compare the performance for quick sample screening we calculated reconstructions using the first ~10 micrographs from each dataset rather than 'cherry picking' the best micrographs. For CTEM we had to use the first 12 micrographs in order to match the particle number of the first 10 VPP micrographs. The ~2000 picked particles from each dataset were subjected directly to a single 3D refinement run in Relion without initial classification, movie processing or particle 'polishing'. The results are shown with triangles in *Figure 6A*. The resolution of the VPP reconstruction was better than the CTEM one by ~ 0.3 Å (3.9 Å vs 4.2 Å). This test shows that VPP may have an advantage over CTEM for initial screening of samples and/or samples containing a mixture of good and bad particles.

Although the 3.2 Å resolution is a remarkable achievement for a phase plate, the performance is still far from what is theoretically expected both in terms of absolute number of particles required to reach a given resolution and in terms of advantage over the conventional approach (*Chang et al., 2010*, Table 2, pol II data). Further practical experience with the phase plate will help to improve the performance. The main factors that currently limit the performance are focusing accuracy, spherical aberration and variation in phase shift. We used a constant amount of defocus for the VPP datasets but a better approach would be to adjust the amount of defocus according to the phase shift. As shown in *Figure 1C* the optimal defocus varies with phase shift. The amount of defocus could be adjusted depending on the number of images already acquired at the current phase plate position. The phase shift versus number of images has to be calibrated in advance, as shown in *Figure 3*. Furthermore, the defocused acquisition approach with a phase plate has to be explored. It is much simpler from a data collection point of view because it does not require precise focusing and has a higher data acquisition throughput, equal to that of CTEM. Future tests are needed to determine which approach is more efficient and to establish if either approach has an advantage for difficult samples.

The VPP matches or slightly exceeds the performance of CTEM for the same number of particles. Because of the high contrast it provides the VPP could be an ideal tool for quick sample screening and/or initial model building. With the in-focus approach the need for precise focusing reduced the image throughput about ~1.7 times (VPP 27 vs CTEM 45 micrographs per hour). Furthermore, setting up VPP experiments is more demanding in terms of user experience. Thus for 'easy' targets, such as relatively big particles and/or such with high symmetry the CTEM can still be more time efficient for achieving high resolutions. In this work we used such a target in order to evaluate the overall performance of the VPP. The full potential of the VPP in terms of improved contrast and uniform spectral coverage will be truly tested and demonstrated by difficult targets, such as small, flexible or heterogeneous samples (*Hall et al., 2011*; *Khoshouei et al., 2016*). We expect that for such targets the extra effort for using the VPP will pay off in terms of getting better reconstructions or a

reconstruction at all. Therefore, the possibilities offered by the VPP could help to widen the target size overlap between cryo-EM and x-ray crystallography.

## Materials and methods

### Sample preparation

*Thermoplasma acidophilum* 20S proteasomes were recombinantly expressed in *Escherichia coli* and purified as described in *Zwickl et al. (1992)*. Samples were plunge-frozen on an FEI Vitrobot Mark III (FEI, Hillsboro, OR). Quantifoil 200 mesh copper R 1.2/1.3 (Quantifoil, Großlöbichau, Germany) holey carbon grids were first cleaned by placing them on a piece of Whatman No. 1 (Whatman, Maidstone, UK) filter paper in a glass Petri dish and then saturating the paper with acetone until the grids were soaked. The Petri dish was left partially open until the acetone evaporated completely. Shortly before plunging the cleaned grids were glow discharged for 30 s in low pressure air in a Harrick plasma cleaner (Harrick, Ithaca, NY). 3 µl of 0.5 mg/ml protein solution was applied on a grid in the Vitrobot chamber set to 95% RH at 20°C then blotted for 5 s and plunged into liquid 37% ethane, 63% propane mixture. Excess cryogen was blotted off from the grid with a piece of filter paper held just above the surface of the liquid nitrogen surrounding the cryogen cup before placing the grid in a plastic cryo grid box for storage.

### Data acquisition

Data was collected on an FEI Titan Krios microscope operated at 300 kV and on an FEI Tecnai F20 microscope operated at 200 kV (FEI, Hillsboro, OR). The FEI Titan Krios was equipped with a Gatan GIF Quantum energy filter, a Gatan K2 Summit direct detector (Gatan, Pleasanton, CA) and an FEI phase plate (FEI, Hillsboro, OR). The acquisition conditions on the FEI Titan Krios were as follows: EFTEM microprobe mode, magnification 37,000x, 50 µm C2 aperture, spot size 6, 1.4 µm beam diameter, zero-loss imaging with 20 eV slit, K2 Summit in counting mode, pixel size 1.35 Å, total dose 30 e$^-$/ Å$^2$, dose rate on the detector 9.1 e$^-$/pixel/s, exposure time 6 s, 12 frames 0.5 s each. The FEI Tecnai F20 was equipped with an FEI Falcon II direct detector, a Gatan 626 cryo-holder and an FEI phase plate. The acquisition conditions on the FEI Tecnai F20 were as follows: nanoprobe mode, magnification 135,000x, 50 µm C2 aperture, spot size 5, pixel size 1.04 Å, total dose 30 e$^-$/ Å$^2$, exposure time 2 s, 7 frames 0.29 s each. On both microscopes we used SerialEM software (*Mastronarde, 2005*) with custom macros for automated single particle data acquisition. For the VPP datasets we used a macro to realize the dual focusing scheme shown in *Figure 2A* with focusing parameters in SerialEM set to: beam tilt angle 10 mrad, focus offset 0, drift protection ON (three image focusing). The focusing mode had identical optical settings as the record mode. The focusing macro measured the defocus on opposite sides of the acquisition area and used the average to correct the defocus. The macro was iterated 3 times to improve the accuracy. To take into account the effect of spherical aberration on the measured defocus ($\triangle z = C_S \beta^2$) 270 nm was added to the target defocus on the FEI Titan Krios (spherical aberration 2.7 mm), i.e. 250 nm target defocus for the -20 nm VPP dataset and 220 nm for the -50 nm VPP dataset. On the FEI Tecnai F20 (spherical aberration 2.1 mm) 210 nm was added, i.e. 190 nm target defocus for the -20 nm VPP dataset. The phase plate was automatically moved to a new position with a 20 µm step every one hour. For the CTEM datasets a single focusing was performed on one side of the acquisition area with drift protection set to OFF in SerialEM to provide more defocus spread. The dataset was collected with a 70 µm objective aperture and a defocus range of -0.8 to -1.7 µm. The acquisition speed was ~27 images/hour for the VPP and ~45 images/hour for the CTEM. The datasets consisted of: Titan Krios VPP −20 nm defocus 200 images, VPP −50 nm defocus 111 images, CTEM 338 images and Tecnai F20 VPP -20 nm defocus 199 images. The phase shift data in *Figure 3* was recorded on the FEI Titan Krios microscope. The optical and acquisition parameters were set to the same values as for the 20S datasets. The beam current was measured by the fluorescent screen of the microscope to be 0.162 nA. Thus the 6 s exposure per image added 0.162 nA x 6 s = 0.97 nC dose to the phase plate. Images were recorded using the same SerialEM macros as for the 20S datasets except the record area was set to be on the carbon film part between the holes of a Quantifoil R 1.2/1.3 plunge-frozen grid. Before each image focusing was performed only once on an adjacent carbon film area with target defocus set to -1.5 µm.

## Data processing

The images and their Fourier transforms were visually inspected and images with visual faults and/or double exposures were rejected. After the initial screening approximately 80% of the original images remained in the datasets: Titan Krios VPP -20 nm defocus 158 images, VPP −50 nm defocus 86 images, CTEM 293 images and Tecnai F20 VPP −20 nm defocus 180 images. The frame stacks were aligned with a homemade GPU accelerated software based on the algorithm of *Li et al. (2013a)*. Particles from the Titan Krios datasets were picked reference-free with the e2boxer program from the EMAN2 software package (*Tang et al., 2007*). Particles from the Tecnai F20 dataset were reference-based picked with the Signature software (*Chen et al., 2007*). The total number of picked particles were: Titan Krios VPP −20 nm defocus 35,469 particles, VPP −50 nm defocus 16,919 particles, CTEM 63,474 particles and Tecnai F20 VPP −20 nm defocus 28,016 particles. To remove false positives and contaminants from each dataset we performed 3D classification runs in Relion 1.4 software (*Scheres, 2012*) with 6 classes and D7 symmetry. The particles from the best looking class were extracted with the numbers being: Titan Krios VPP -20 nm defocus 13,469 particles (38%), VPP -50 nm defocus 6,340 particles (38%), CTEM 35,717 particles (56%) and Tecnai F20 VPP -20 nm defocus 16,145 particles (58%). Those particles were then used to perform a 3D refinement in Relion followed by movie processing, particle 'polishing' and another 3D refinement of the 'polished' particles. The B-factors reported by Relion after post-processing of the final 3D maps were: Titan Krios VPP −20 nm defocus −122.6, VPP -50 nm defocus -124, CTEM -119.3 and FEI Tecnai F20 VPP −20 nm defocus -330.4. In order to calculate the resolution versus number of particles data in *Figure 6* random subsets with varying number of particles were extracted from the Titan Krios VPP −20 nm and CTEM datasets. Each subset was subjected to a complete 3D refinement run with a 60 Å low-pass filtered initial model.

## Acknowledgements

We are grateful to Jan Schuller for kindly providing the purified 20S proteasome sample. We thank Jürgen Plitzko for his technical support.

## Additional information

### Competing interests

RD: Co-inventor in US patent US9129774 B2 "Method of using a phase plate in a transmission electron microscope". WB: Scientific Advisory Board of FEI Company.

### Funding

| Funder | Author |
| --- | --- |
| Max-Planck-Gesellschaft | Wolfgang Baumeister |

The funders had no role in study design, data collection and interpretation, or the decision to submit the work for publication.

### Author contributions

RD, Conception and design, Acquisition of data, Analysis and interpretation of data, Drafting or revising the article; WB, Conception and design, Drafting or revising the article

### Author ORCIDs

Radostin Danev, http://orcid.org/0000-0001-6406-8993

## Additional files

### Major datasets

The following datasets were generated:

| Author(s) | Year | Dataset title | Dataset URL | Database, license, and accessibility information |
|---|---|---|---|---|
| Danev R, Baumeister W | 2015 | Volta phase plate in-focus dataset of T20S proteasome | http://www.ebi.ac.uk/pdbe/emdb/empiar/entry/10057 | Publicly available at EMPIAR (Accession no: EMPIAR-10057) |
| Danev R, Baumeister W | 2015 | Cryo-EM dataset of T20S proteasome | http://www.ebi.ac.uk/pdbe/emdb/empiar/entry/10058 | Publicly available at EMPIAR (Accession no: EMPIAR-10058) |
| Danev R, Baumeister W | 2015 | Volta phase plate 3.2A resolution reconstruction of the T20S proteasome | http://www.ebi.ac.uk/pdbe/entry/emdb/EMD-3347 | Publicly available at the EBI Protein Data Bank in Europe (Accession no: EMD-3347). |
| Danev R, Baumeister W | 2015 | cryo-EM structure of T20S proteasome | http://www.ebi.ac.uk/pdbe/entry/emdb/EMD-3348 | Publicly available at the EBI Protein Data Bank in Europe (Accession no: EMD-3348). |

The following previously published dataset was used:

| Author(s) | Year | Dataset title | Dataset URL | Database, license, and accessibility information |
|---|---|---|---|---|
| Campbell MG, Veesler D, Cheng A, Potter CS, Carragher B | 2015 | 2.8 Angstrom resolution reconstruction of the T20S proteasome | http://www.ebi.ac.uk/pdbe/entry/emdb/EMD-6287 | Publicly available at the EBI Protein Data Bank in Europe (Accession no: EMD-6287). |

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
