## [Decision Letter]

Thank you for submitting your work entitled "Cryo-EM single particle analysis with the Volta phase plate" for consideration by *eLife*. Your article has been favorably evaluated by Michael Marletta (Senior editor) and three reviewers – Raimon Ravelli, Robert Glaeser, and Sjors Scheres, who is a member of our Board of Reviewing Editors.

The reviewers have discussed the reviews with one another and the Reviewing Editor has drafted this decision to help you prepare a revised submission.

Summary:

This manuscript reports a very significant advance in single-particle cryo-EM methodology, namely use of an electron-optical phase-shifting device to produce high-resolution, in-focus images of weak-phase objects. The specimen consisted of 20 S proteasomes, and the use of single-particle data-processing resulted in a three-dimensional reconstruction at a resolution of 3.2 Å. Although the authors produced a similar result from images obtained by defocusing the image, this work is still exceptionally innovative and important because it demonstrates that the phase plate does not introduce some unforeseen factor that limits the resolution. With that issue now settled, future work is expected to show whether the higher contrast, provided when using a phase plate, is able to significantly extend the capabilities of single-particle cryo-EM, especially in the case of structures much smaller than the proteasome. Therefore, all three reviewers heartily recommend publication in *eLife*.

Essential revisions:

1) The 2015 PNAS paper of Danev et al. provide hard data for beam-induced phase shift of a function of dose (nC) for a specific VPP. In the Results and Discussion section the authors write that the "phase plate versus number of images has to be calibrated in advance", like they did in the PNAS paper. Assuming that this calibration is solely a function of dose and not of time (e.g., is there any decay of the phase shift during the time the data acquisition of the next hole is being setup?), such calibration could have been done with very little effort. Such data are important for the reproducibility of their own experiments as well as for future work done in other labs: could they still be added?

2) Figure 5: one should plot the (natural) logarithm of the number of particles against 1/d^2^, the inverse of the square of the resolution. That way, one can fit straight lines through the red and blue curves to obtain B-factors, which can then be compared directly.

3) The smaller subsets in Figure 5 were obtained based on a sorting (in RELION) of the particle images. This should be replaced by random subsets of varying size, as sorting the best particles into the smallest subsets may have the undesired effect that the resolution for the smaller subsets is relatively better.

---

## [Author Response]

*Essential revisions: 1) The 2015 PNAS paper of Danev et al. provide hard data for beam-induced phase shift of a function of dose (nC) for a specific VPP. In the Results and Discussion section the authors write that the "phase plate versus number of images has to be calibrated in advance", like they did in the PNAS paper. Assuming that this calibration is solely a function of dose and not of time (e.g., is there any decay of the phase shift during the time the data acquisition of the next hole is being setup?), such calibration could have been done with very little effort. Such data are important for the reproducibility of their own experiments as well as for future work done in other labs: could they still be added?*

We measured and added phase shift versus image number/total dose data as Figure 3 in the revised manuscript.

2) Figure 5: one should plot the (natural) logarithm of the number of particles against 1/d^2^, the inverse of the square of the resolution. That way, one can fit straight lines through the red and blue curves to obtain B-factors, which can then be compared directly. 3) The smaller subsets in Figure 5 were obtained based on a sorting (in RELION) of the particle images. This should be replaced by random subsets of varying size, as sorting the best particles into the smallest subsets may have the undesired effect that the resolution for the smaller subsets is relatively better.

We recalculated the resolution versus particle number data using random subsets of particles and replaces the original data. The results are plotted in Figure 6 in the revised manuscript. We also plotted the ln(N) vs 1/d^2^ in Figure 6 and fitted linear regions of the data to measure the B-factor. As described in the text we observed two regions with approximately twofold change in the B-factor. Unfortunately we could not provide a plausible explanation for the variation of the B-factor.